# Modulation of protective reflex cough by acute immune driven inflammation of lower airways in anesthetized rabbits

Laurent Foucaud[1], Bruno Demoulin[1], Anne-Laure Leblanc[1], Iulia Ioan[1,2‡], Cyril Schweitzer[1,2‡], Silvia Demoulin-Alexikova[1,2] *

**1** Research Unit EA 3450 DevAH—Development, Adaptation and Handicap, Campus Biologie Santé, University of Lorraine, Vandœuvre-Lès-Nancy, France, **2** Department of Pediatric Functional Testing, Hôpital d'Enfants, CHRU de Nancy, Vandoeuvre-Les-Nancy, France

☯ These authors contributed equally to this work.
‡ These authors also contributed equally to this work.
* silvia.demoulin-alexikova@univ-lorraine.fr

**Data Availability Statement:** All relevant data are within the manuscript and its Supporting Information files.

## Abstract

Chronic irritating cough in patients with allergic disorders may reflect behavioral or reflex response that is inappropriately matched to the stimulus present in the respiratory tract. Such dysregulated response is likely caused by sensory nerve damage driven by allergic mediators leading to cough hypersensitivity. Some indirect findings suggest that even acid-sensitive, capsaicin-insensitive A-δ fibers called "cough receptors" that are likely responsible for protective reflex cough may be modulated through immune driven inflammation. The aim of this study was to find out whether protective reflex cough is altered during acute allergic airway inflammation in rabbits sensitized to ovalbumin. In order to evaluate the effect of such inflammation exclusively on protective reflex cough, C-fiber mediated cough was silenced using general anesthesia. Cough provocation using citric acid inhalation and mechanical stimulation of trachea was realized in 16 ovalbumin (OVA) sensitized, anesthetized and tracheotomised rabbits 24h after OVA (OVA group, n = 9) or saline challenge (control group, n = 7). Number of coughs provoked by citric acid inhalation did not differ between OVA and control group (12,2 ±6,1 vs. 17,9 ± 6,9; p = 0.5). Allergic airway inflammation induced significant modulation of cough threshold (CT) to mechanical stimulus. Mechanically induced cough reflex in OVA group was either up-regulated (subgroup named "responders" CT: 50 msec (50–50); n = 5 p = 0.003) or down-regulated (subgroup named "non responders", CT: 1200 msec (1200–1200); n = 4 p = 0.001) when compared to control group (CT: 150 msec (75–525)). These results advocate that allergen may induce longer lasting changes of reflex cough pathway, leading to its up- or down-regulation. These findings may be of interest as they suggest that effective therapies for chronic cough in allergic patients should target sensitized component of both, reflex and behavioral cough.

**Funding:** This work was supported by Ministry of High Education and Research of France (Ministère de l'Enseignement supérieur et de la Recherche) under contrat EA3450 DevAH. http://www.enseignementsup-recherche.gouv.fr/. The funders had no role in study design, data collection and analysis, decision to publish, or preparation of the manuscript.

**Competing interests:** The authors have declared that no competing interests exist.

## Introduction

A common feature to all allergic and atopic disorders is the presence of strong sensations, and/or reflexes that are primary aimed to get rid of allergen [1]. However, mediators released during an allergic reaction interact with afferent nerves and may induce longer-lasting changes in neuronal activity that can outlast the stimulus by hours, days or even years [1]. As nervous system is rendered hyperactive in many patients with allergic diseases, symptoms such as chronic itch, pain, gastrointestinal discomfort, dyspnea, sneezing, coughing or wheezing frequently represent dysregulated sensory responses and pathological behaviors driven by neuroimmune interaction. [1,2]. This concept is in keeping with a hypothesis of neuropathic origin of chronic cough that have been raised recently highlighting the possible cause-effect link between exposure to infectious, irritant or allergic diseases and sensory nerve damage resulting in cough hypersensitivity [3].

Recent studies focusing on neurobiology of cough [4,5,6,7] reveal the complexity of cough regulation resulting in reconfiguration of brainstem respiratory pattern generator in such a way that normal respiratory pattern is replaced by cough motor pattern [8,9]

To summarize current knowledge, cough motor pattern may be produced reflexively or behaviorally. Reflex cough has a crucial role in basic defense of airways including clearance of aspirate, inhaled particulate matter and accumulated secretions [5,10,11]. Reflex cough is largely dependent upon brainstem processing of incoming afferent input and requires minimal conscious involvement and it can be therefore provoked in anesthetized animals and humans or in decerebrated animals. On the other hand, behavioral cough occurs as a result of subcortical and cortical processing of afferent information from airways. It is associated with conscious perception of the sensation of irritation (urge to cough) and serves *in fine* to reduce such sensation. [7,10,12].

Stimulation of the acid-sensitive, capsaicin-insensitive mechanoreceptors projecting to the larynx, trachea and large bronchi leads to coughing even in anesthetized animals and humans and is therefore likely responsible for protective reflex cough [5,10,11]. This type of mechanoreceptors classed as A-δ fibres whose soma is situated in nodose ganglia were recently called "cough receptors" [5].

Chemically sensitive bronchopulmonary non-myelinated C-fibers, whose soma is situated in jugular ganglia, mediate cough evoked by a wide range of nociceptive stimuli, including endogenous mediators released during inflammation or tissue injury through expression of Transient Receptor Potential (TRP) ion channels on their nerve terminals. Inhalation of C-fiber stimulators, such as capsaicin, induces not only cough, but also a sensation of urge-to-cough; which is often experienced prior to the cough motor event. Thus, it remains possible that bronchopulmonary C-fibers do not evoke reflex cough, but rather promote behavioral cough [11,12]. Studies in experimental animals may support this notion since capsaicin is extremely effective at evoking cough in conscious animal but typically fail to do so in anesthetized one [13,14].

Based on the evidence to date, chronic irritating cough in patients with allergic disorders may therefore reflect either behavioral or reflex response that is inappropriately matched to the stimulus present in the respiratory tract [10]. In order to treat chronic cough in such patients, it is essential to identify possible mechanisms of both, reflex and behavioral cough sensitization, driven by allergic mediators. This task appears especially complicated as new therapeutic approaches should not suppress reflex cough, that protects respiratory system from aspiration, inhaled particulate matter and clears the airways form accumulated secretions [15], but rather restore normal reflex cough function and sensitivity [16].

TRP channels frequently act as a final pathway for many immune cell-derived stimuli in order to activate or modulate sensory nerves. Therefore, the type of afferent neuron initiating cough that is most susceptible to direct activation and modulation by allergic mediators is bronchopulmonary C-fibre [17]. For this reason, in order to develop new therapeutic strategies much attention is payed to understand mechanisms of its activation. As TRP receptors are not normally expressed by nodose A-δ fiber [5,14,18,19], one may expect that these afferent nerves are not activated by allergen mediators and so, the protective reflex cough is not altered during allergic inflammation.

However, many indirect findings suggest that immune-driven inflammation modulates also reflex cough. Several endogenous mediators that are released by immune or structural cells during different stages of allergic disease may influence afferent and central neurons involved in cough reflex pathway. [11]. [20,21],

The aim of this study is to find out whether reflex cough is altered during acute allergic airway inflammation in a rabbit model sensitized to ovalbumin. The used of rabbit model to study neuronal mechanisms implied in cough reflex induction to different stimuli has been advocated for neuronal similarity of rabbit lungs to human lungs [22,23,24]. In order to evaluate the effect of such inflammation exclusively on reflex cough supposed to be mediated by A-δ fibers, C-fiber mediated cough was silenced using general anesthesia. As "cough receptors" are sensitive to punctuate mechanical stimulation of epithelium and a rapid drop in luminal pH, the effect of airway allergic inflammation was tested for the both stimuli. The use of validated and reproducible methodology of mechanical stimulation of trachea, elaborated in our laboratory permitted us to assess the sensitivity of cough reflex to mechanical stimulation by using several mechanical stimulation durations. We also analyzed bronchoalveolar lavage fluid for immune cell counts to characterize the inflammatory state.

We addressed the hypothesis that citric acid induced cough reflex and mechanically induced cough reflex in anesthetized animals are up-regulated 24h after OVA challenge in sensitized rabbits.

## Materials and methods

### Animals

Sixteen New Zealand adult rabbits (1.5–2 kg) purchased by HYCOLE (SARL-HYCOLE-Route de Villers Plouich, 59159 MARCOING,France, http://hycole.com/) were studied. All animals were housed. All animals were housed in two in conventional animal facilities with a 16 hour day and 8 hour night cycle at Animal House of Faculty of Medicine, University of Lorraine. Food and drink were given *ad libitum* and routinely checked by the technical staff. Enrichment consisted in hay and small pieces of wood. Animal care and study protocol was approved by the local Ethics Committee on animal testing (Comité d'éthique en expérimentation animale CEEA) affiliated to the University of Lorraine (Comité d'Ethique Lorrain en Matière d'Experimentation animale CELMEA C2EA-66) followed by the validation of the "Ministère de l'Enseignement Supérieur et de la Recherche" under the number authorization 01582.02 according to recommendations 86–609 CEE issued by the council of the European Communities. Ethics Committees on animal testing are recognized as the competent authority for the ethical evaluation of authorization applications for projects involving animal models. The CEEA comprise at least 5 members: a veterinary surgeon, a scientist, an investigator, an animal keeper, and a person from the social profession not involved in research activities. In 2016, 126 CEEA were authorized by the French Ministry of Research, including that of University of Lorraine.

## Sensitization and provocation of airway inflammation with ovalbumin

The sensitization and challenge protocol used in this study is shown in Fig 1. All rabbits were sensitized by two intraperitoneal injections (IP) of 1 mL of a suspension containing 0.1 mg ovalbumin (Sigma-Aldrich, Saint Quentin Fallavier, France) and 10 mg aluminum hydroxide (Sigma-Aldrich, Saint Quentin Fallavier, France) in sterile saline (0.9% NaCl) on day 0 and 13 [25]. The solutions were suspended by 30min agitation on ice before IP injections. Seven days before airway exposure to allergen, the sensitization to OVA was tested by an intradermal sensitization assay. One hundred microliters of OVA solution (200 µg/mL of saline) and saline alone were injected symmetrically in the shaved back dermis of each animal using a 1 mL syringe fitted with a 25-gauge needle. The extent of the reaction was measured 24h and 48 h later using caliper. Two measurements were taken at right angles to one another in order to calculate the induration size in square millimeters [26]. According to the size of skin induration around the site of OVA injection, rabbits were divided into Saline and OVA group i.e. challenged by Saline or OVA aerosols, respectively. Rabbits with smallest skin indurations were intended to Saline group which corresponded to 7 controls versus 9 OVA.

The aerosols of ovalbumin or saline were performed using an ultrasonic nebulizer (LS 290, SYSTAM®) producing droplets with a mass media aerodynamic diameter of 3.5 µm (60% of particles have a size between 1 and 5 µm) connecting to custom-built device elaborated in our laboratory. This system was a closed Plexiglas chamber (33x33x51.5 cm) consisting of a head chamber isolated from a body chamber. The rabbit was positioned in order to permit proximity between rabbit nose and an opening for aerosol diffusion. To realize the aerosol challenges, rabbits were placed in the restraining system that had the opening for tubing connected to nebulizer containing 20 mL of 2.5 mg/mL of OVA in saline or saline alone. Twenty-minute nebulization allowed aerosolization of almost 40 mg of OVA. All aerosols were performed in class I safety enclosure for safety reasons. Rabbits were exposed to OVA or saline, 48h and 24h before the mechanical and chemical cough challenge. A habituation to aerosol challenges was performed for five days (ten minutes each challenge) before the first OVA/Saline challenge.

## Chemical and mechanical stimulation of trachea

**Anesthesia, analgesia and euthanasia.** Anesthesia was induced with sodium pentobarbital (22 mg.Kg-1) (Ceva Santé animale, Libourne, France) injected through the ear vein. Fifteen minutes after the induction of anesthesia an intramuscular injection in hind limb muscles of

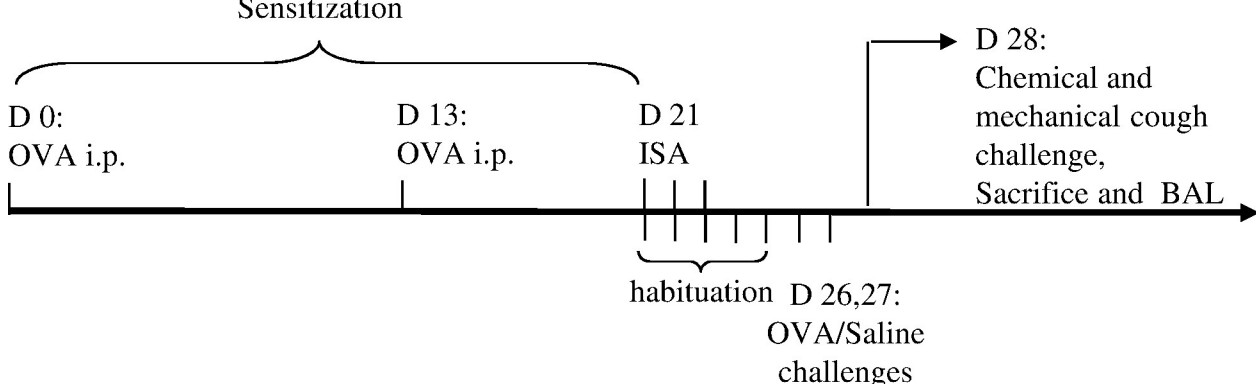

**Fig 1. Experimental procedure.** IP: intraperitoneal injections of 0.1 mg ovalbumin (OVA) and 10 mg aluminum hydroxide, ISA: Intradermal Sensitization Assay with 20 µg of OVA or saline, OVA challenges:40 mg of OVA, BAL: Bronchoalveolar Lavage.

ketamine (10 mg. $Kg^{-1}$) (Virbac France, Carros, France) was realized. The anesthesia was maintained by a bolus injection of sodium pentobarbital (2.2 mg. $Kg^{-1}$) following by a continuous perfusion of 33 mg/hour. Monitoring of analgesia and anesthetic depth was performed in 15 minutes intervals by assessing the change of physiologic parameters (heart and respiratory rate) and absence of withdrawal responses to compressive ear and toe pinch. In the case of withdrawal response, a supplementary bolus of sodium pentobarbital (2.2 mg. $Kg^{-1}$) was injected. In this case, the procedure was interrupted for 15 min.

At the end of the experiment, animals were sacrificed by an overdose of pentobarbital.

**Animal preparation.**  The anesthetized animal was placed in the supine position. Rectal temperature was continuously monitored with an electrical thermistor (Physitemp Instruments, YSI 402 Clifton, NJ USA) and maintained at 38°C using circulating warm water pad. The electromyographic activity of the *rectus abdominis* muscle was measured by insertion of bipolar insulated fine stainless steel wire electrodes (A-M Systems INC, Sequim, WA 98,382) introduced under visual control into either the transversus abdominis or external oblique abdominal muscles according to Basmajian and Stecko ([27] to further characterize the active expiration from augmented breath [28]. An upper cervical tracheotomy allowed the insertion of a tracheal cannula that was connected to a pneumotachograph (No. 0 Fleisch pneumotachograph with linear range ± 250 mL/s) and to the mechanical stimulation apparatus. The pneumotachograph was calibrated before each experiment using a 20 ml calibration syringe.

**Mechanical stimulation of trachea.**  The apparatus developed to elicit a discrete mechanical challenge to the trachea has been described in details and validated in previous reports [29]. Briefly, a rotating silastic catheter introduced in tracheotomy is driven by a small electrical motor that spins the catheter and rubs its tip onto the airway mucosa for a short period. The electrical signal from the engines serves as marker for accurate identification of the stimulus time course. As cough reflex is significantly more frequently provoked during inspiration compared to expiration [29] mechanical stimulations were triggered during inspiratory phase. The beginning of inspiration was detected electronically as soon as the flow signal reached a positive value. Four stimulation durations (50, 150, 300 and 600 msec) repeated 3 times each in a pseudorandom order were performed. An interval of at least 1 min of quiet and regular breathing was allowed to elapse between two stimuli. during which reference tidal volume was determined [30].

**Chemical stimulation of trachea.**  A 1M citric acid (Sigma-Aldrich, Saint Quentin Fallavier, France) solution was nebulized using an ultrasonic nebulizer (LS 290, SYSTAM) in a plastic bottle connected to the tracheal cannula for 4 min.

## Bronchoalveolar lavage (BAL)

Bronchoalveolar lavage was performed following chemical and mechanical stimulation of trachea. After euthanasia a catheter was inserted *via* steel tracheal cannula into trachea until a resistance prevented to progress. Lavage was performed by slow injection followed by aspiration of 5 mL of HEPES (140 mM NaCl, 5 mM KCl, 1 $MgCl_2$, 10mM glucose, and 10 mM HEPES; pH 7.4) solution for three times. The collected solution (10–13 mL) was homogenized by successive aspiration before filtration on nylon tissue with 60 μm mesh to remove mucus (Zschauer et al., 1999). The total cell concentration of BAL was determined after trypan blue coloration on counting chamber Malassez. An aliquot of the BAL containing 250,000 cells was then spun in a cytospin (500 rpm for 10 min) and stained by May-Grünwald-Giemsa (MGG) staining to identify cell populations using light microscopy (40x objective, AX70 Olympus®). A minimum of 100 cells per slide were identified and classified as macrophages, eosinophils, lymphocytes, basophils, neutrophils, monocytes, epithelial cells and other cells.

## Data analysis

**Ventilatory responses to mechanical stimulation of trachea.**   The cough response was identified from the change in tidal volume (VT), peak expiratory flow (V'Epeak) and *rectus abdominis* EMG [28]. Cough reflex (CR) was defined as an increase of VT followed by an increased V'Epeak associated with a

burst of rectus abdominis EMG activity (Fig 2). In order to take into account the spontaneous between-breath variability, an unbiased differentiation of CR from expiration reflex was achieved by a statistical evaluation of VT between stimulation and reference breath. Tidal volume of reference breath was determined as the mean of 3 breaths prior to stimulation and its upper limit as mean + 3 standard deviations. The cough reflex was identified when VT of stimulation breath was higher than upper limit of reference VT. Reflex cough response to mechanical stimulation consisted of a bout of one or several CR.

The use of four stimulation durations (50, 150, 300 and 600 msec) allowed the assessment of duration–response curve enabling to assess the sensitivity of cough response to mechanical

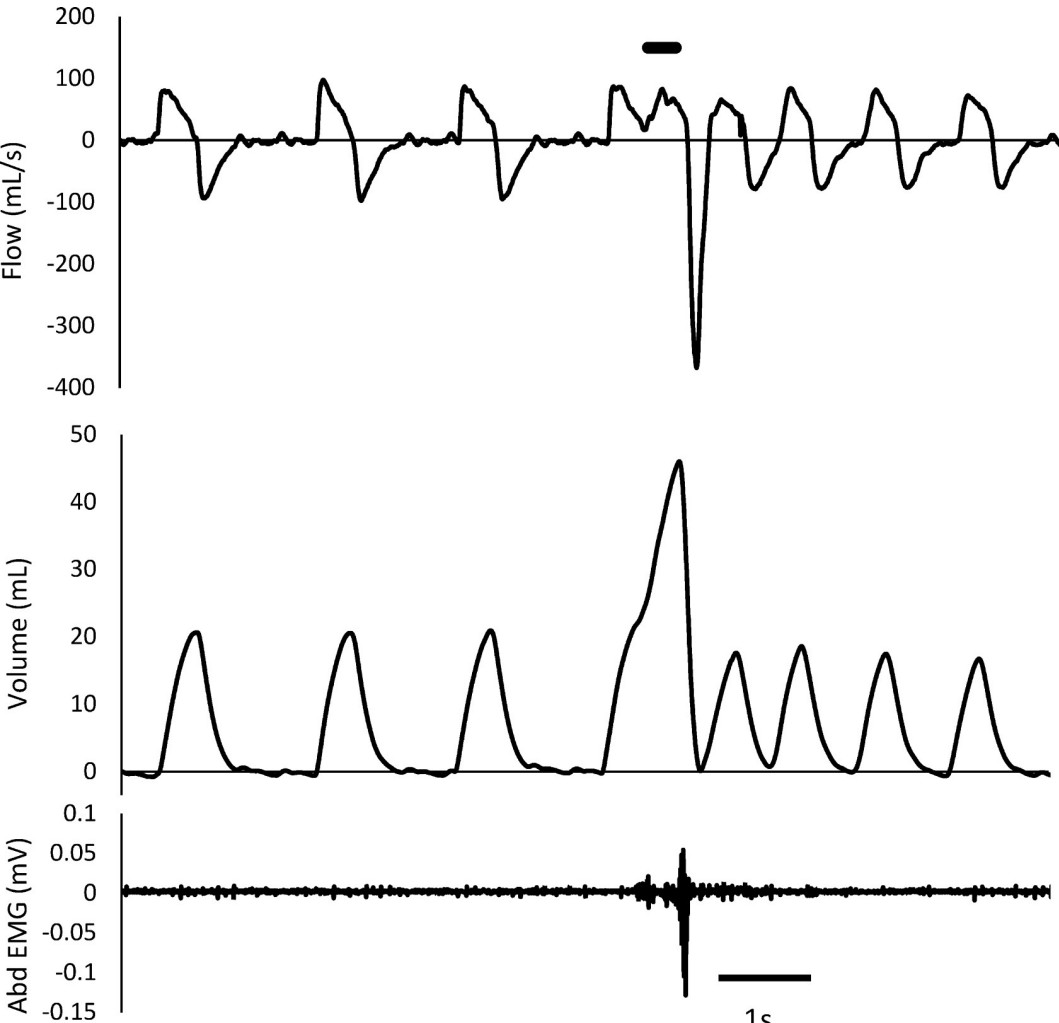

**Fig 2. Typical cough reflex to mechanical stimulation of trachea.** The increased expiratory flow is associated with a burst of *rectus abdominis* EMG activity and preceded by an increased tidal volume. Thick bar on the top indicates the moment of mechanical stimulation.

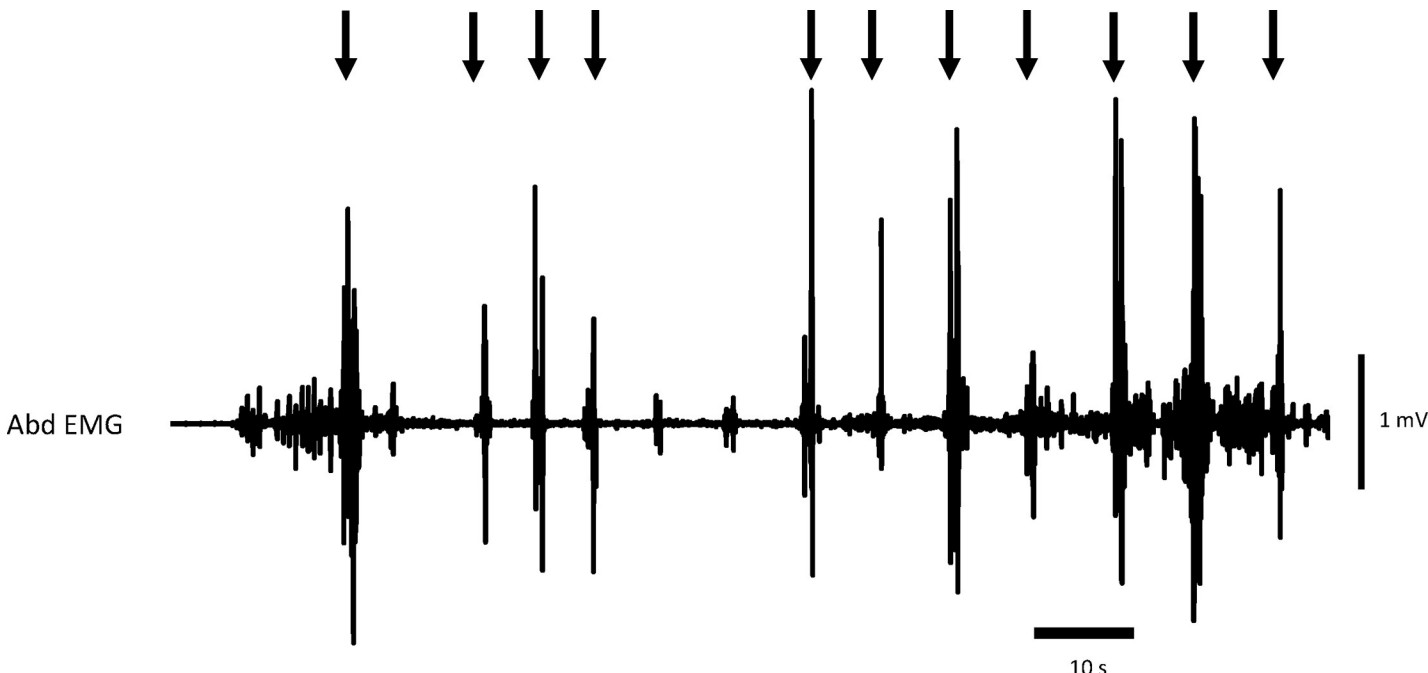

**Fig 3. Typical reflex cough response during chemical stimulation of trachea quantified from the bursts of the *rectus abdominis* EMG activity.** Downward arrows indicate the bursts of EMG activity corresponding to cough reflex.

stimulation. The cough threshold (CT) was defined as the shortest stimulation duration necessary to provoke at least one cough reflex. When no response was seen using 600 msec (longest) stimulation, the threshold was arbitrarily set to 1200 msec.

In addition, the cumulative number of CR for each stimulation duration (NCUM50, NCUM150, NCUM300, NCUM 600, respectively) was calculated as the sum of cough reflex number induced by that duration and all shorter stimulation durations.

**Ventilatory responses to chemical stimulation of trachea.** Cough response to nebulisation of citric acid for 4 minutes was quantified from the bursts of the *rectus abdominis* EMG activity that allowed to count number of CR (Fig 3). The cumulative number of CR at 1st (NCUM1), 2nd (NCUM2), 3rd (NCUM3) and 4th minute (NCUM4) were calculated.

## Statistical analysis

Statistical analysis was performed using the SYSTAT 12 package (San Jose, CA, USA). Cells types' number, cumulative number and total number of defensive responses were expressed as the mean ± standard error and statistical comparisons between groups were performed using ANOVA or Student's t test.

Threshold of defensive response was expressed as median (25–75%) and statistical comparisons between groups were performed using non parametric Kruskal-Wallis test. Pairwise comparisons between groups were subsequently performed using post-hoc analysis (Conover-Inman test for multiple comparisons) and a statistical significance was retained for a $p \leq 0.05$.

## Results

### Effect of allergen challenge on airway eosinophilia

Rabbits exposed to OVA aerosols showed a significant increase of eosinophil counts in BAL as compared to those exposed to Saline (11.6±1.92.0% vs. 2.2±0.6%; p = 0.001) (Fig 4.)

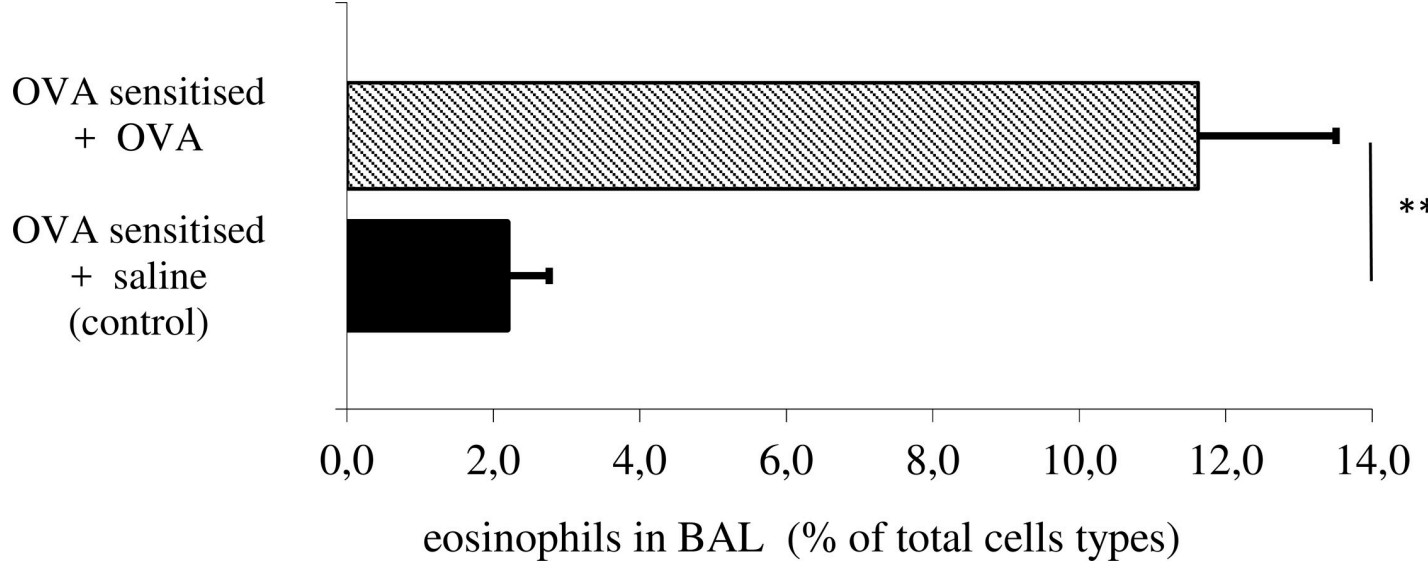

**Fig 4. Eosinophil count in bronchoalveolar lavage (BAL) of Saline and OVA group at the end of experiment.** Eosinophil count expressed as percentage of total cell count. Data are mean ± SE. P-values were calculated using the unpaired, two-tailed Student's t-test.

### Effect of allergen challenge on cough response to airway stimulation

**Cough response to chemical stimulation.** Saline and OVA group did not statistically differ neither in total number of CR (= NCUM4) provoked by nebulization of citric acid aerosol during 4 minutes (saline: 17,9 ± 6,9 vs. OVA: 12,2 ±6,1; p = 0.5) nor in NCUM1, NCUM2, NCUM3 (Fig 5).

**Cough response to mechanical stimulation.** Mechanical stimulation of trachea in rabbits challenged with OVA provoked two diametrically different responses (Fig 6). In order to obtain appropriate understanding of reflex cough modulation by airway inflammation, OVA group was therefore divided into 2 subgroups for further analysis.

The subgroup named "non responders" consisted of 5 rabbits (5/9; 55.5%) that did not cough at any of four stimulation durations. In this subgroup CT was significantly increased compared to saline group (Table 1) and NCUM600 was significantly lower compared to Saline group (Table 1, Fig 7).

The subgroup named "responders" consisted from 4 rabbits (4/9; 44.5%) presenting a cough response when using the shortest mechanical stimulation. In this subgroup, CT was significantly decreased compared to saline group (Table 1) and NCUM50, NCUM300 and NCUM600 were significantly higher compared to saline group (Table 1; Fig 7).

The cough response to citric acid stimulation and the eosinophil counts in BAL were not significantly different between the two OVA subgroups (Table 1).

## Discussion

This study is the first one to investigate the effect of acute immune-driven inflammation associated with allergic challenge on reflex cough in sensitized rabbits *in vivo*. Cough response to citric acid inhalation was not altered by two consecutive allergen challenges performed 48h and 24h before reflex cough provocation when compared to animals challenged by saline. Unlike citric acid induced cough, allergic airway inflammation induced significant modulation of mechanically induced reflex cough that was either up-regulated or down-regulated when compared to rabbits challenged by saline.

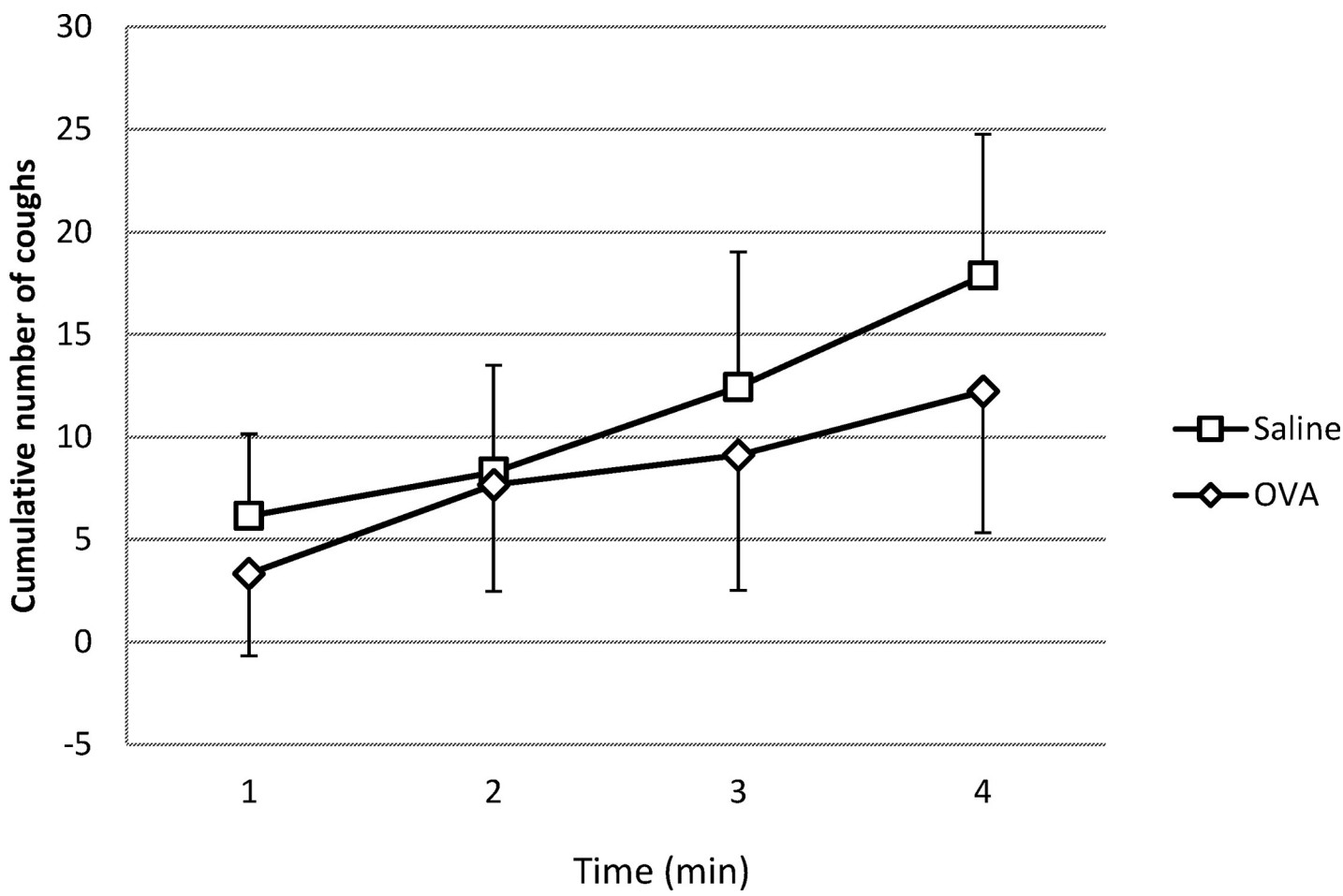

**Fig 5. Cumulative number of coughs provoked by 4 minute nebulization of citric acid aerosol (1M).** Data are mean ± SEM. Cumulative number of coughs did not differ significantly between Saline and OVA.

These findings are of interest as they show that protective reflex cough, an important defense mechanism that allow clearing airways even during sleep or anesthesia, is modulated during allergic inflammation of lower airways.

### Immune-driven inflammation model

In order to evaluate modulation of involuntary cough during acute allergic airway inflammation we have employed the frequently used allergen OVA. Acute sensitization of OVA followed by OVA aerosol challenges induces allergic airway and pulmonary inflammation in many animal models of asthma, including rabbit [31,32,33]. Although, this procedure not entirely reflect the inflammation profile of asthma [32], increased eosinophils in BAL and lung tissue have been attested in different animals models *i.e.* rabbit [34], mouse [35] and guinea pig [36]. Recent literature shows that rabbit model is of particular interest to study asthma and other lung diseases for several reasons [23] and we used this model especially for two reasons. Firstly, mouse and rat, the most frequently used models of asthma, do not express typical cough reflex (at least under anesthesia) and secondly, in guinea pig, the most frequently used animal model of cough, this reflex cannot be provoked under pentobarbital anaesthesia (personal observations). The protocol of sensitization was modified to take into consideration both

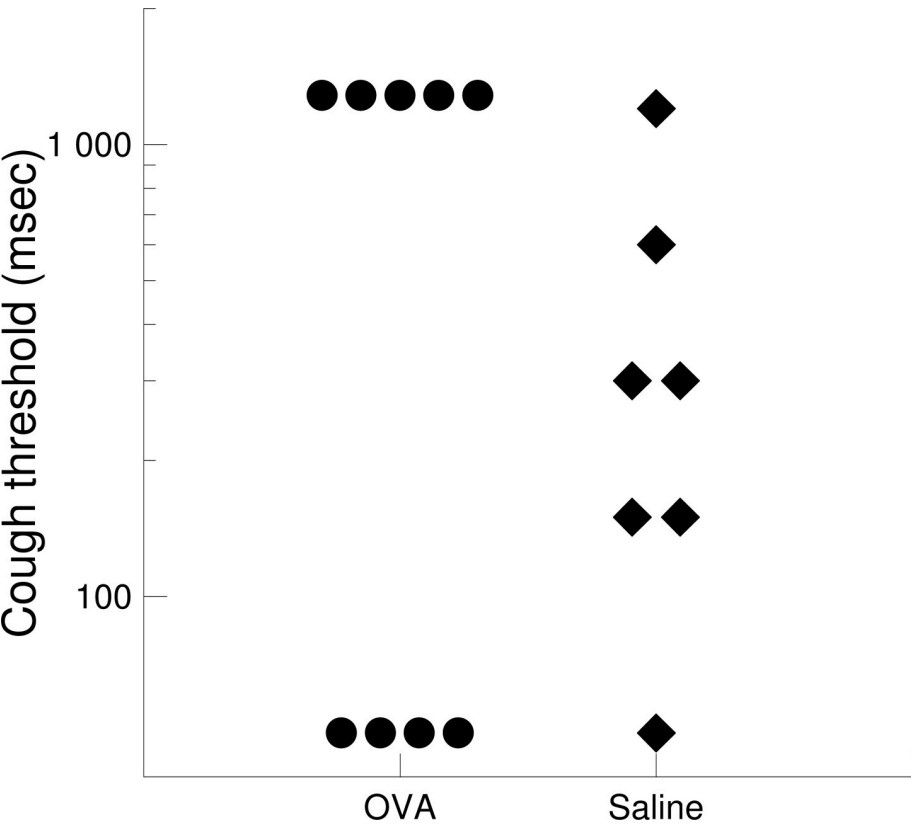

**Fig 6. Scatter plot displaying the values of mechanical cough threshold in rabbits of OVA and Saline group.**
Cough threshold represents the shortest stimulation duration necessary to provoke at least one cough reflex. When no response was seen using longest (600 msec) stimulation, the threshold was arbitrarily set to 1200 msec.

the efficiency of inflammation induction and the 3Rs principle in animal experimentation. Indeed, in the study of Zschauer et al. (1999) [31] the same profile of cell was observed between

**Table 1. Parameters of cough response to mechanical stimulation and to chemical stimulation and BAL cell count.** Comparison of cough threshold to mechanical stimulation (CT) and the cumulative number of cough for each stimulation duration (NCUM50, NCUM150, NCUM300, NCUM 600) between two subgroups of OVA group and Saline group.

| | OVA | | Saline | p | |
|---|---|---|---|---|---|
| | **Non responders** | **Responders** | | **Non responders vs. Saline** | **Responders vs. Saline** |
| **n** | **5** | **4** | **7** | | |
| **Cough response to mechanical stimulation of trachea** | | | | | |
| CT | 1200 (1200–1200) | 50 (50–50) | 150 (75–525) | **0.001** | **0.003** |
| NCUM 50 | 0.00±0.24 | 1.75±0.26 | 0.14±0.2 | 0.89 | **0.001** |
| NCUM 150 | 0.00±0.74 | 3.75±0.83 | 1.14±0.6 | 0.48 | 0.06 |
| NCUM 300 | 0.00±1.11 | 7.00±1.24 | 2.57±0.63 | 0.22 | **0.03** |
| NCUM 600 | 0.00±1.26 | 9.75±1.4 | 4.57±1.06 | **0.04** | **0.03** |
| **Cough response to chemical stimulation of trachea (citric acid)** | | | | | |
| Total count | 11,8 ± 4,8 | 12,75±5,5 | 17,8±6.9 | 0.85 | 1.0 |
| **Bronchoalveolar lavage (% of total cell count)** | | | | | |
| Eosinophils | 11.3 ± 2.4 | 12.0 ± 3.3 | 2.36 ± 0.6 | **0.012** | **0.017** |
| Macrophages | 86.1 ± 1.6 | 86.1 ± 4.0 | 96.26 ±0.85 | **0.006** | **0.012** |

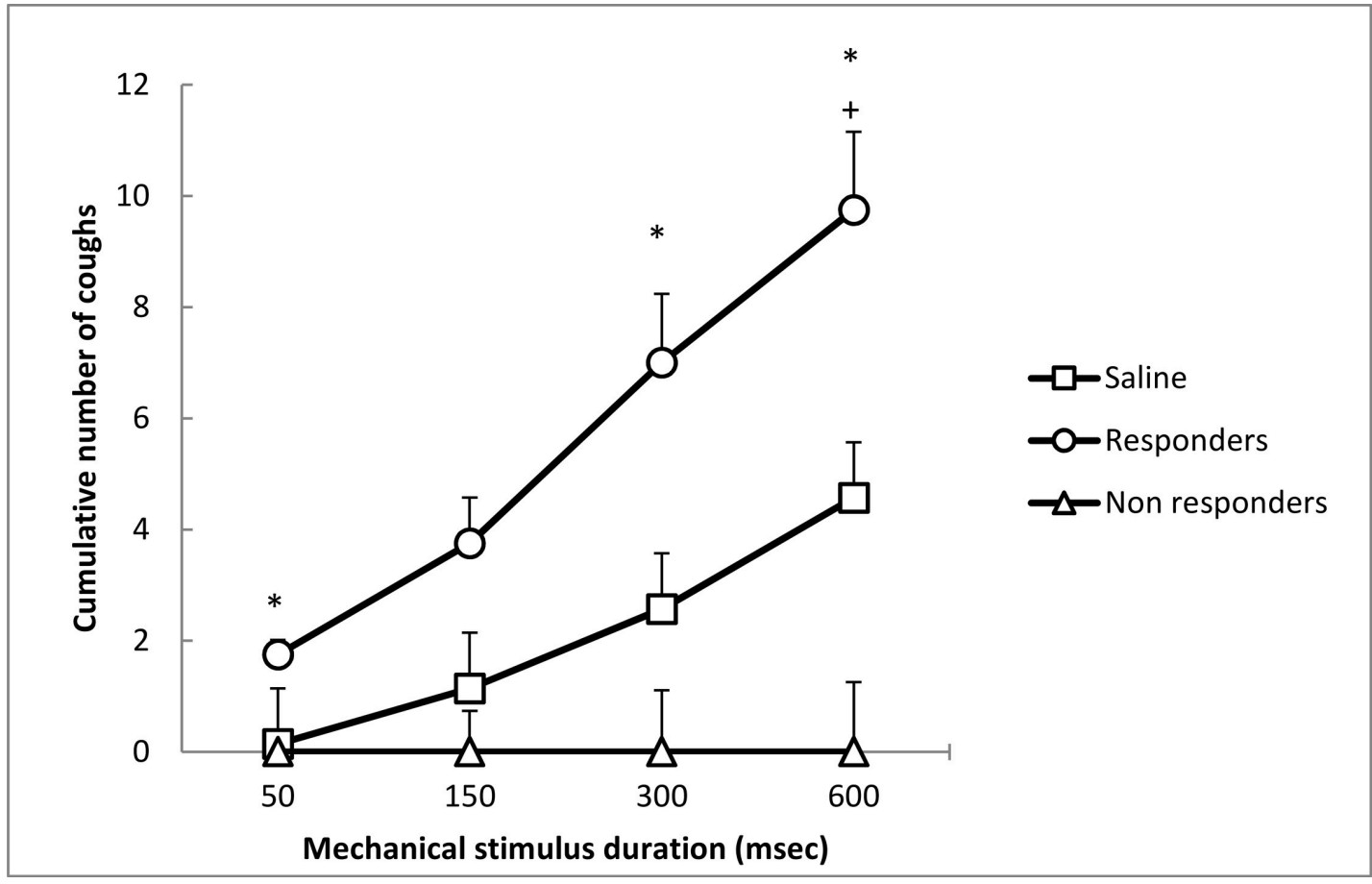

**Fig 7. The cumulative number of coughs provoked by mechanical stimulation in Saline group and two OVA subgroups (Responders and Non responders)**
*Significance of difference between "Responders" and Saline group at the level of p < 0.05; + Significance of difference between "Non responders" and Saline group at the level of p < 0.05.

control rabbits (no-sensitized and saline challenged) and sensitized rabbits challenged with OVA. As variability in sensitization efficacy was suspected, it has been chosen to reduce the number of animals used by a systematic sensitization. To test the efficiency of sensitization a dermal test was realized and as far as possible, rabbits with smallest indurations were intended to control group i.e. not exposed to OVA aerosols (S1 Fig and S1 Table). As the use of OVA aerosol leads to the development of an antigen tolerance and the quick resolution of inflammation after discontinuing antigen exposures [32] we realized preliminary experiments to determine the optimal aerosol challenge number that allowed us to reproduce some key features of clinical acute asthma, such as early and late-phase bronchoconstriction in response to allergen challenge and the influx of eosinophils into lower respiratory tract resulting in eosinophil airway inflammation. Realization of two OVA challenges in sensitized rabbits, 48 h and 24 h before experiments, provoked significant increase of eosinophils (S1 Table) that was comparable with those in study of Xue and co-workers that used five allergen challenges [37]. The timing of eosinophils influx after OVA challenge was studied in mouse and revealed discrepancy corresponding to maximum inflammations from 6h [35] to 24h [38] post allergen exposure. This discrepancy was explained by the type of allergen. In rabbit, a 24 h delay after the last aerosol exposure appears as the common procedure.

## Modulation of chemically induced cough

The first hypothesis addressed in our study expected that protective reflex cough provoked by citric acid will be up-regulated in sensitized animals exposed to allergen when compared to those exposed to control solution. This prediction was based on prior studies that describe several potential mechanisms by which allergic airway inflammation may modulate neural pathways responsible for cough reflex. Firstly, citric acid-induced cough in anesthetized animals without airway inflammation is supposed to be exclusively mediated through ASICS channels expressed on airway Aδ-fiber terminals [11]. Up to date, there is no direct experimental evidence suggesting that ASICS channels expressed on airway Aδ-fiber terminals are modulated by allergic inflammation. However, it has been shown that ASIC transcript levels in sensory neurons were increased in inflammatory conditions *in vivo* [39] and that proinflammatory mediators such as nitric oxide [40], nerve growth factor and in a lesser extent serotonin [41] increases proton-gated currents on sensory neurons and/or the number of ASIC expressing neurons leading to a higher sensory neuron excitability. Secondly, there is experimental evidence suggesting that increased production of various neurotrophic factors during airway allergic inflammation may lead to phenotypic changes in neurons [42,43,44]. For example, Lieu and co-workers have found that nodose A-δ neurons in guinea pig trachea begin to express *de novo* TRPV1 channels during allergic airway inflammation [42]. As TRPV1 channels are involved in citric acid-evoked cough, de novo expression of TRPV1 channels on cough receptor cell membrane may *in fine* lead to increased responsiveness of these fibers to acid solutions. Finally, there is considerable evidence that following allergen challenge the non-myelinated and myelinated afferent neurons of respiratory tract begin to synthetize and transport tachykinins to their central terminal [20,21]. As the local release of substance P has been shown to increase the excitability of the nucleus of the solitary tract neurons in guinea pigs and rabbits [45,46], allergic inflammation may lower the activation threshold of cough reflex evoked by cough receptor stimulants via the mechanisms of tachykinin-mediated central sensitization [1]. However, our results showed no difference of cough reflex number provoked by citric acid between sensitized animals exposed to OVA and those exposed to saline (Fig 3). Our results do not highlight cough reflex up-regulation 24 hours after the last OVA challenge in rabbits challenged daily for 2 consecutive days. It is therefore supposed that a) the reflex cough provoked by citric acid is not up-regulated by airway inflammation; or b) the number of allergen expositions in our study was not sufficient to induce neuromodulation of afferent and/or second order neurons; and/or finally c) the time needed to induce such neuromodulation after allergen challenge was insufficient. The latter two suggestions seem to be more likely even if the acute inflammation state was attested by eosinophils influx in BAL. The *de novo* expression of TRPV1 channels on vagal myelinated fibers was observed in guinea pigs challenged daily for 3 consecutive days [42] and in rats challenged 3 times per week for 3 weeks [43] and increased excitability of brainstem neurons was induced by a long-term exposure of young rhesus monkeys to house dust mite [47]. More studies are needed to clarify the time course of inflammation mediated modulation of reflex cough induced by chemical stimulation.

## Modulation of mechanically induced cough

Mechanical stimulation of lower airways is converted to electrical signals via depolarization of afferent vagal Aβ-fibers, Aδ-fibers or C-fibers [11,48]. Among them, the low-threshold mechanosensitive nodose Aδ fibers that terminate in extra pulmonary airways have been recently defined as "cough receptors" in the airways of guinea pig [14]. Different experimental studies focusing on anatomical and physiological characteristics of the Aδ fibers have shown

that these afferents, exquisitely sensitive to light touch are responsible for mediating mechanically induced cough reflex [11]. Our hypothesis of mechanical cough reflex up-regulation by allergic airway inflammation was based on several observations. In the *in vitro* study realized by Riccio and co-workers (1996) the mechanical sensitivity of tracheal Aδ-fiber endings to calibrated Frey filaments was increased immediately after 5–30 min exposure to antigen challenge when compared to pre-challenge period [48]. Given the similarities between pain and cough, the results showing that allergic stimulus may produce mechanical hyperalgesia lasting approximately 72 hours suggest that allergic challenge may induce a longer-lasting increase in cough reflex sensitivity to mechanical stimulus. Results of our study partially confirmed such expectations as they showed that significantly increased cough reflex sensitivity to mechanical stimulus was present 24h after allergen challenge in almost half of OVA treated rabbits.

On the other hand, in resting half of OVA treated rabbits, we found significantly down-regulated cough response when compared to control group. Transduction of mechanical sensing is not clarified yet, however, there is a lot of information supporting the role of cytoskeleton and/or extracellular matrix. Concerning respiratory tract, *in vitro* electrophysiological study, focusing to the mechanisms of adaptation of airway afferent neurons to mechanical stimulation, suggested that nodose mechanosensitive fibers are localized in the tissue in a fashion that the activation stimulus is the dynamic phase of the mechanical stimulus [49]. Using new techniques of intravital labeling with FM2-10 and visualization through water-immersion optics confirmed adherence of Aδ-fiber terminals to the sub epithelial matrix, where sensing physical disturbances is signaled through the structure of the matrix [50].

It is therefore not surprising that responsiveness to mechanical stimulation may be modulated in the case of airway wall modifications driven by acute allergic response.

Acute exposition to allergen in sensitized subjects is commonly associated with modifications of airway wall and mucus composition. Exudation of plasma proteins from leaky blood vessels is frequently observed in this situation and leads to development of rapid perivascular edema through the action of inflammatory mediators [51,52]. Inflammatory edema of tracheal wall may considerably change the structure of ECM by making it stiffer. If afferent neurons embedded in ECM experience mechanical stimulation when ECM undergoes large deformations, it is tempting to suppose that increased rigidity of ECM might increase the intensity of mechanical stimulus needed to provoke cough. In addition, acute allergic inflammatory response of the airway includes important modifications of mucus secretion. In human, the conducting airways of healthy subjects are covered by thin mucous layer containing 97% of water and few mucins [53]. In acute asthma, an extensive mucins secretion [54,55] and an impaired degradation of mucus in the presence of plasma proteins exudation [51] results in formation of concentrated mucus of rubbery quality [56]. An increase of mucus viscosity might increase the intensity of mechanical stimulus needed to provoke cough. Therefore, a combination of inflammatory edema and qualitative changes of mucous layer might participate on the no response pattern observed in this study.

In the study of Riccio et al. the mechanical sensitivity of tracheal Aδ-fiber endings to calibrated Frey filaments was measured in vitro using a preparation consisting from isolated tracheal segments, *vagus* nerves and afferent ganglions [48]. In such case, exudation of plasma proteins, inflammatory edema and subsequent increase of mucus viscosity could not take place and the increase of mechanical sensitivity of tracheal Aδ-fibers in this study probably occurred in absence of airway tracheal wall modifications.

Taken together, the two responsiveness profiles to mechanical stimulation found in our study likely reflect the relation between mechanotransduction and the time course of the inflammation process and recovery if we suppose individual phenotypic responses to allergen challenge in different animals. Unfortunately, there is no information about spontaneous

recovery time from tracheal edema provoked by acute inhalation of allergen. There is some information showing that the restoration of the mucus rheology proprieties in subjects with acute asthma exacerbation takes place 24–72 hours after the initiation of the exacerbation [51]. Therefore, we suppose that animals with down-regulated mechanical cough response did not yet recover from inflammatory edema and mucus modifications whereas those with up-regulated cough response did. Such speculation seems likely as our recent observations showed that mechanical cough was significantly down-regulated immediately after single nebulization of OVA in sensitized rabbits and such down-regulation remained during the totality of recording period (100 min post-challenge) [57].

To confirm such relation an investigation of mucus composition and histological tissue analysis should be realized in each animal, however, such studies were not in the goal of this work.

Further studies are necessary to test these hypotheses and to focus on the impact of the morphological modifications induced by inflammation on mechanically induced cough reflex mediated by Aδ-fiber.

### Limitations of the study

This study presents several limitations.

The evaluation of inflammatory state was only realized by eosinophils count in BAL without further histological characterization of the airways. Such supplementary investigations would have been useful to better define inflammation induced modifications of airway tissue which likely modify the potential of intratracheal stimulation.

During citric acid cough challenge, it was not possible to record airflow using pneumotachograph and so the fine differentiation of cough from expiration reflex (ER) could not be realized. Expiration reflex is frequently elicited by mechanical stimulation of larynx and trachea [58] and so its differentiation from cough is important for this modality. On the other hand, there is no information whether citric acid inhalation may provoke ER as the two reflexes are commonly not differentiated in the studies using citric acid to provoke cough. Only one study, using citric acid microinjections into the larynx, classified expulsive responses into CR or ER [59]. In their study, microinjections of 0.8 M citric acid during 5 minutes provoked a big cough response (about 26–30 coughs) but only few ER ($n \approx 3$) and no dose response relationship was observed for ER when using several citric acid concentrations. Therefore, we suppose that in our study, the great majority of expulsive events provoked by inhalation of citric acid and quantified from the bursts of the *rectus abdominis* EMG activity were cough reflexes.

The number of animals per group was determined between 6 and 9 animals during the design of this study. The analysis of results revealed that the OVA group included two distinct profiles responses to mechanical cough stimulation. It has been chosen to distinguish two subgroups named "non responders" consisted of 5 rabbits (5/9; 55.5%) and "responders" consisted of 4 rabbits (4/9; 44.5%). For this reason, this study is a pilot study and further studies will be realised to determine the airways changes in the two subgroups.

### Conclusion and perspectives

The effect of acute allergic airway inflammation on protective reflex cough sensitization was studied in anaesthetized animal in order to silence C-fiber mediated cough. The results of this study suggest that modulation of mechanically induced reflex cough is present 24 hours after allergen challenge in sensitized animals. On the other hand, this study failed to highlight the modulation of citric acid induced reflex cough in the same setting.

These results advocate that allergen may induce longer lasting changes of protective reflex cough pathway, leading to its up- or down-regulation. Further studies are needed, however, to understand underlying mechanisms of reflex cough modulation driven by allergic mediators so as their time course.

These findings may be of interest as they suggest that effective therapies for chronic cough in allergic patients should target sensitized component of both, reflex and behavioral cough.

The rabbit model of reflex cough testing presented here may prove useful in evaluating the reflex cough function and sensitivity when developing new therapies for chronic cough.

## Supporting information

**S1 Fig. Sensitization test to ovalbumin.** Sensitization was induced by two intraperitoneal injections of OVA (200 μg/mL) in 2 weeks intervals. Dermal test was performed by two symmetrical intradermic injections of OVA (20 μg in 100 μL of saline) and of saline (NaCl 0.9%) one week after the last injection. Induration sizes (A) and erythema (B) were evaluated 24h and 48h after intradermic injections.
(DOCX)

**S1 Table. Percentage of macrophages and eosinophils in BAL of rabbits following different OVA exposure procedures. Preliminary results before study.** Results are expressed as the mean ± SD of number of counts indicated in brackets. The number of rabbits of the different groups is indicated in column entitled n.
(DOCX)

## Acknowledgments

The authors gratefully acknowledge Professor François Marchal and François Plénat, for their expert advices to conduct this investigation. The authors acknowledge also D. Meng for her helpful assistance in histological techniques.

## Author Contributions

**Conceptualization:** Laurent Foucaud, Silvia Demoulin-Alexikova.

**Formal analysis:** Laurent Foucaud, Bruno Demoulin, Silvia Demoulin-Alexikova.

**Investigation:** Laurent Foucaud, Bruno Demoulin, Anne-Laure Leblanc, Silvia Demoulin-Alexikova.

**Methodology:** Laurent Foucaud, Bruno Demoulin, Silvia Demoulin-Alexikova.

**Project administration:** Laurent Foucaud, Silvia Demoulin-Alexikova.

**Resources:** Bruno Demoulin, Anne-Laure Leblanc.

**Software:** Bruno Demoulin.

**Supervision:** Laurent Foucaud, Silvia Demoulin-Alexikova.

**Validation:** Laurent Foucaud, Iulia Ioan, Cyril Schweitzer, Silvia Demoulin-Alexikova.

**Visualization:** Laurent Foucaud, Bruno Demoulin, Silvia Demoulin-Alexikova.

**Writing – original draft:** Laurent Foucaud, Bruno Demoulin, Silvia Demoulin-Alexikova.

**Writing – review & editing:** Laurent Foucaud, Iulia Ioan, Cyril Schweitzer, Silvia Demoulin-Alexikova.

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
