## [Decision Letter · Decision Letter 0]

30 Sep 2019

PONE-D-19-22841

Modulation of protective reflex cough by acute immune driven inflammation of lower airways in anesthetized rabbits

PLOS ONE

Dear Dr. Demoulin-Alexikova,

Thank you for submitting your manuscript to PLOS ONE. After careful consideration, we feel that it has merit but does not fully meet PLOS ONE’s publication criteria as it currently stands. Therefore, we invite you to submit a revised version of the manuscript that addresses the points raised during the review process.

Both reviewers raised some minor issues that require to be addressed. I hope that the authors can effectively respond to these comments in the revision.

We would appreciate receiving your revised manuscript by Nov 14 2019 11:59PM. To enhance the reproducibility of your results, we recommend that if applicable you deposit your laboratory protocols in protocols.io, where a protocol can be assigned its own identifier (DOI) such that it can be cited independently in the future. For instructions see: http://journals.plos.org/plosone/s/submission-guidelines#loc-laboratory-protocols

We look forward to receiving your revised manuscript.

Kind regards,

Yu Ru Kou, PhD

Academic Editor

PLOS ONE

Journal Requirements:

2. At this time, we request that you  please report additional details in your Methods section regarding animal care, as per our editorial guidelines:

(1) Please state whether the provided ethics committee contains animal welfare experts or whether an animal ethics or IACUC committee reviewed and approved the study. Please provide the full name of the committee that reviewed and approved the study  

(2) Please state the source of the rabbits used in the study  

(3) Please provide details of animal welfare (e.g., shelter, food, water, environmental enrichment)

3. Please state specifically whether the IRB approved your study.

Additional Editor Comments (if provided):

Reviewers' comments:

Reviewer's Responses to Questions

**Comments to the Author**

1. Is the manuscript technically sound, and do the data support the conclusions?

Reviewer #1: Yes

Reviewer #2: Yes

2. Has the statistical analysis been performed appropriately and rigorously? 

Reviewer #1: Yes

Reviewer #2: Yes

3. Have the authors made all data underlying the findings in their manuscript fully available?

Reviewer #1: Yes

Reviewer #2: Yes

4. Is the manuscript presented in an intelligible fashion and written in standard English?

Reviewer #1: Yes

Reviewer #2: Yes

5. Review Comments to the Author

Reviewer #1: I was very satisfied by the manuscript, because it showed the role of airway Ad fibers - minly their contribution to the modulatin of airway defensive reflexes by allergen challenge. Surprisingly there were no changes in the citric acid induced cough and mechanical challenge provoked two brand different responses - no response or up regulated response, which authors try to explain by the presence of mucus/fluid layer and therefore insensitivity of mechanosensory afferents. I think this study should be published as a pilot study with further experiments focused on the airway changes in any, and neurobiology. i do not think 24 or 48 hours lasting eposure to inflammatory mediators could cause "neuroplastic" changes suffiecient to upregulate coughing mediated via Ad. Anyway, it will be interesting to see how this model will behave after repeated challenges ..saying a chronic model with analysis of ion channel expression.

I have a question for author. Do you think 5 and 4 animals in responders /nonresponders group - is this enough to be ready to jump to conclusions? A paragrpah should be devoted to the number of subjects in your experiment.

Introduction section should be a little shorter. I see that authors were trying to explain the situation that exist in this field precisely, but to my taste it was too long. a Part of it perhaps should be moved to discussion - but this is on the journal policy if they preffer longer introductions due to a broader spectrum of readers.

Reviewer #2: Interesting paper about the influence of allergic inflammation induced by sensibilisation and challenge by ovalbumin in rabbits on cough. I appreciate logic sequence of methods, results and discussion part and find the results as very interesting..

Minor rcomments and recommendation:

1. I recommend to modify an abstract in order to be clear which procedures were made in the two groups.

2. Is there any reason why non-sesnitized rabbits were not used as a control?

3. Reasons for using challenge by ovalbumin only in "positive" dermal tests after sensitisation should be included in the study - what was the size of induration for inclusion into respective goups?

4. It is questionalble to use results of own unpublished observations in this kind of study; on the other hand, they help to elucidate observed effects.

5. Were there any differences in BAL eosinophils between responders and non-responders in OVA challenged group? This and other markers of inflammation could help to explain these completely copntrary results in cough threshold (fig. 6)... Did you measure also some other systemic markers of allergic inflammation in plasma (e.g. some interleukins)? If yes, woud you consider their inclusion into the manuscript to clearly demonstrate also systemic changes?

6. In the introduction, a paragraphs about suitability of rabbit model for testing both chemical and mechanical induced cough should be included with appropriate references.

7. When performing BAL, the volume of HEPES solution repeatedly admnistered is usually re-calculated to body weight of animals. Why you did adinister uniform amount of the solution? Was the bw of rabbith equally distrobuted among the groups?

8. Latin words should be written in Italics (e.g. rectus abdominis).

9. In the discussion should be mentioned, why only sensitised animals where used and were not confirmed with non-sensitized animals (e.g. eosinophils).

10. The first sentence in the discussion should be more specific for rabbits, as there are available some studies about ovalbumin sensitisation and challenge on cough reflex in guinea pigs (https://www.ncbi.nlm.nih.gov/pubmed/18204154;
https://www.ncbi.nlm.nih.gov/pubmed/12470206;
https://www.ncbi.nlm.nih.gov/pubmed/21939521;
https://www.ncbi.nlm.nih.gov/pubmed/16343041;
https://www.ncbi.nlm.nih.gov/pubmed/15611590;
https://www.ncbi.nlm.nih.gov/pubmed/15569475;
https://www.ncbi.nlm.nih.gov/pubmed/19218639)

6. PLOS authors have the option to publish the peer review history of their article (what does this mean?). If published, this will include your full peer review and any attached files.

Reviewer #1: No

Reviewer #2: No

---

## [Author Response · Author response to Decision Letter 0]

12 Nov 2019

Response to Reviewers

Journal Requirements:

Answer: 

Some modifications have been done in the formatting of the manuscript:

“Abstract” title size was increase to the appropriate size

Numbers in front of the affiliations was in superscript

The all text format was checked to be conform to the PLOS ONE style requirements

2. At this time, we request that you please report additional details in your Methods section regarding animal care, as per our editorial guidelines:

(1) Please state whether the provided ethics committee contains animal welfare experts or whether an animal ethics or IACUC committee reviewed and approved the study. Please provide the full name of the committee that reviewed and approved the study 

Answer:

Section “animals” in material and methods has been amended as follow 

Animal care and study protocol was approved by the local ethics committee affiliated to the University of Lorraine (Comité d’Ethique Lorrain en Matière d’Experimentation animale CELMEA C2EA-66) followed by the validation of the “Ministère de l’Enseignement Supérieur et de la Recherche” under the number authorization 01582.02 according to recommendations 86-609 CEE issued by the council of the European Communities. Animal care and health were under supervision by the “Services Vétérinaires Départementales de Meurthe et Moselle”.

(2) Please state the source of the rabbits used in the study 

Answer:

Section “animals” in material and methods has been amended as follow 

Sixteen New Zealand adult rabbits (1.5 -2 kg) purchased by HYCOLE (SARL-HYCOLE-Route de Villers Plouich , 59159 MARCOING,France,http://hycole.com/) were studied .

(3) Please provide details of animal welfare (e.g., shelter, food, water, environmental enrichment)

Answer:

Section “animals” in material and methods has been amended as follow 

Sixteen New Zealand adult rabbits (1.5 -2 kg) purchased by HYCOLE (SARL-HYCOLE-Route de Villers Plouich , 59159 MARCOING,France,http://hycole.com/)were studied . All animals were housed in two in conventional animal facilities with a 16 hour day and 8 hour night cycle at Animal House of Faculty of Medicine, University of Lorraine. Food and drink were given ad libitum and routinely checked by the technical staff. Enrichment consisted in hay and small pieces of wood. ….. 

3. Please state specifically whether the IRB approved your study.

Answer:

Section “animals” in material and methods has been amended as follow 

….Animal care and study protocol was approved by the local ethics committee affiliated to the university of Lorraine (Comité d’Ethique Lorrain en Matière d’Experimentation animale CELMEA) followed by the validation of “Ministère de l’Enseignement Supérieur et de la Recherche” under the number authorization 01582.02 according to recommendations 86-609 CEE issued by the council of the European Communities. Animal care and health were under supervision by the “Services Vétérinaires Départementales de Meurthe et Moselle”.

Reviewer #1 :

1. Do you think 5 and 4 animals in responders /non responders group - is this enough to be ready to jump to conclusions? A paragraph should be devoted to the number of subjects in your experiment.

Answer:

During the design of the study the number of animals per group was determined between 6 and 9 animals. The initial hypothesis was to induce an increase of cough reflex to mechanical stimulation in the OVA group. The analysis of results revealed that the OVA group included two distinct profiles responses to mechanical cough stimulation. It has been chosen to distinguish two subgroups named “non responders” consisted of 5 rabbits (5/9; 55.5%) and “responders consisted of 4 rabbits (4/9;44.5%). As mentioned, this study is a pilot study and further studies will be realised to determine the airways changes in the two subgroups.

Section “limitations of the study” has been amended as follow 

Limitations of the study

….The number of animals per group was determined between 6 and 9 animals during the design of this study. The analysis of results revealed that the OVA group included two distinct profiles responses to mechanical cough stimulation. It has been chosen to distinguish two subgroups named “non responders” consisted of 5 rabbits (5/9; 55.5%) and “responders” consisted of 4 rabbits (4/9;44.5%). For this reason, this study is a pilot study and further studies will be realised to determine the airways changes in the two subgroups.

2. Introduction section should be a little shorter. I see that authors were trying to explain the situation that exist in this field precisely, but to my taste it was too long. a Part of it perhaps should be moved to discussion - but this is on the journal policy if they prefer longer introductions due to a broader spectrum of readers.

Answer:

 We have shortened Introduction section as proposed by the reviewer. 

Reviewer #2:

1. I recommend to modify an abstract in order to be clear which procedures were made in the two groups.

Answer:

Abstract has been amended as follow 

Mechanically induced cough reflex in OVA group was either up-regulated (subgroup named “responders” CT: 50 msec (50 – 50); n = 5 p=0.003) or down-regulated (subgroup named “non responders”, CT: 1200 msec (1200 - 1200); n=4 p=0.001) when compared to control group (CT: 150 msec ( 75 - 525)).

2. Is there any reason why non-sensitized rabbits were not used as a control?

Answer:

In accordance with the 3Rs principle in animal experimentation, it has been chosen to reduce the total number of animals in the study. We have therefore chosen to use sensitized but not exposed rabbits as a control and to omit the use of non-sensitized rabbits as another control. Brief revision of published studies using OVA to model allergic inflammation have revealed that majority of studies used as a control group either non-sensitized animals or sensitized but not exposed animals, rarely both control group types. As can be seen from the figure below, in current study, responses of sensitized rabbits to dermal tests showed a relatively broad induration size range (see S1 figure below). Such different intensity response to dermal test was supposed to reflect a variability of sensitization efficacy. As far as possible, rabbits with smallest indurations were intended to control group (not exposed to OVA aerosols) and those with largest indurations to OVA group (exposed to OVA aerosols), which corresponded to 7 controls versus 9 OVA. Given such variability of sensitisation efficacy, our approach seems to be adequate for the aim of our study. Moreover, cough threshold observed in control group of current study (150 msec (75 - 525 msec) was similar to that observed in another study performed by our laboratory and using identical protocol of mechanical cough provocation (300 msec (150 - 600 msec) (Varechova, Demoulin et al. 2015). This suggests that cough threshold is not affected by OVA sensitisation. 

S1 Figure: Sensitization test to ovalbumin.

Sensitization was induced by two intraperitoneal injections of OVA (200 µg/mL) in 2 weeks intervals. Sensitization evaluation was performed by two symmetrical intradermic injections of OVA (20 µg in 100 µL of saline) and of saline (NaCl 0.9%) one week after the last injection. Induration sizes (A) and erythema (B) were evaluated 24h and 48h after intradermic injections.

A

B

 24h 48 h

Erythema NaCl 

0.9% OVA 200µg/mL NaCl 

0.9% OVA 200µg/mL

none 16 9 12 7

observed 0 7 0 5

Reference: Varechova, S., B. Demoulin, A. L. Leblanc, L. Coutier, I. Ioan, C. Bonabel, C. Schweitzer and F. Marchal (2015). "Neonatal hyperoxia up regulates cough reflex in young rabbits." Respir Physiol Neurobiol 208: 51-56.

Discussion was modified as follows:

….The study protocol was modified to take into consideration both the 3Rs principle in animal experimentation and the efficiency of inflammation induction. In accordance with the 3Rs principle, it has been chosen to reduce the total number of animals in the study. We have therefore used only sensitized but not exposed rabbits as a control and omitted the use of non-sensitized rabbits as another control. In the study of Zschauer et al. (1999) [31] the same profile of cell was observed between control rabbits (not sensitized and saline challenged) and sensitized rabbits challenged with OVA. Moreover, cough threshold observed in control group of current study (150 msec (75 - 525 msec) was similar to that observed in another study using identical protocol of mechanical cough provocation (300 msec (150 - 600 msec) (Varechova, Demoulin et al. 2015). Responses of sensitized rabbits to dermal tests showed a relatively broad induration size range and as far as possible, rabbits with smallest indurations were intended to control group i.e. not exposed to OVA aerosols and those with largest indurations to OVA group (exposed to OVA aerosols) (S1 Fig and S1 Table)…

3. Reasons for using challenge by ovalbumin only in "positive" dermal tests after sensitisation should be included in the study - what was the size of induration for inclusion into respective groups?

Answer:

As mentioned above, as far as possible the rabbits with smallest indurations were intended to control group. The cut off size of induration was determined as 400 mm2. The allocation conducted to a control group with a mean wheal area of 216 mm2 ± 173 mm2 and a OVA group with a mean wheal of 799 mm2 ± 453 mm2. Three rabbits with induration under 400 mm2 were included in this last group included for practical reasons.

4. It is questionable to use results of own unpublished observations in this kind of study; on the other hand, they help to elucidate observed effects.

Answer:

The results mentioned in the discussion and referred as “unpublish results” (see after) have been presented in the ERS international congress 2017, 13-15 September in Milan, Italy. (Foucaud L et al., Up-regulation of chemically induced but not mechanically induced cough during early-phase response to allergen inhalation in ovalbumin sensitized rabbits). The citation of the abstract has been included in the references. 

Discussion has been amended as follow 

Such speculation seems likely as our recent observations showed that mechanical cough 

was significantly down-regulated immediately after single nebulization of OVA in sensitized rabbits and such down-regulation remained during the totality of recording period (100 min post-challenge) [59]. 

Following reference was added in reference list:

59. Silvia Demoulin-Alexikova, Laurent Foucaud, Bruno Demoulin, Anne-Laure Leblanc, Iulia Ioan, François Marchal, Cyril Schweitzer. Up-regulation of chemically-induced but not mechanically-induced cough during early-phase response to allergen inhalation in ovalbumin sensitized rabbits. European Respiratory Journal 50 (suppl 61) PA1149; DOI: 10.1183/1393003.congress-2017.PA1149 Published 6 December 2017

5. Were there any differences in BAL eosinophils between responders and non-responders in OVA challenged group? This and other markers of inflammation could help to explain these completely contrary results in cough threshold (fig. 6)... Did you measure also some other systemic markers of allergic inflammation in plasma (e.g. some interleukins)? If yes, would you consider their inclusion into the manuscript to clearly demonstrate also systemic changes?

Answer:

No difference was observed in BAL eosinophils between responders and non-responders as mentioned in the table 1, see extract below

Table 1. extract Parameters of cough response to mechanical stimulation and to chemical stimulation and BAL cell count. 

 OVA Saline p

 Non responders Responders Non responders vs. Saline Responders vs. Saline

n 5 4 7 

Bronchoalveolar lavage (% of total cell count)

Eosinophils 12.3 ± 2.83 12.34 ± 3.47 2.36 ± 0.6 0.012 0.017

Macrophages 85.30 ± 2.05 85.75 ± 4.06 96.26 ±0.85 0.006 0.012

The measurement of plasmatic and BAL allergic inflammatory parameters (interleukins) other that eosinophils were planned in the future studies focused on the mechanistic exploration of the preliminary result presented in this work. 

6. In the introduction, a paragraph about suitability of rabbit model for testing both chemical and mechanical induced cough should be included with appropriate references.

Answer:

The first reviewer suggested that the introduction is perhaps a bet long. So, to take into account this first remark and to answer the question, the modification in introduction was voluntarily as short as possible but give appropriate references justifying the rabbit model for this study.

Introduction was modified as follow

The aim of this study is to find out whether reflex cough is altered during acute allergic airway inflammation in a rabbit model sensitized to ovalbumin. The used of rabbit model to study neuronal mechanisms implied in cough reflex induction to different stimuli has been advocated for neuronal similarity of rabbit lungs to human lungs (Spina et al., 1998; Keir and Page 2008; Clay et al., 2016). In order to evaluate the effect of such inflammation exclusively on reflex cough supposed to be mediated by A-δ fibers, C-fiber mediated cough was silenced using general anesthesia. As “cough receptors” are sensitive to punctuate mechanical stimulation of epithelium and a rapid drop in luminal pH, the effect of airway allergic inflammation was tested for the both stimuli. The use of validated and reproducible methodology of mechanical stimulation of trachea, elaborated in our laboratory permitted us to assess the sensitivity of cough reflex to mechanical stimulation by using several mechanical stimulation durations.

References: 

1. Spina D, Matera GM, Riccio MM, Page CP. A comparison of sensory nerve function in human, guinea-pig, rabbit and marmoset airways. Life Sci. 1998;63: 1629–1642. 

2. Keir S, Page C. The rabbit as a model to study asthma and other lung diseases. Pulm Pharmacol Ther. 2008;21: 721–730. doi:10.1016/j.pupt.2008.01.005

3. Clay E, Patacchini R, Trevisani M, Preti D, Branà MP, Spina D, et al. Ozone-Induced Hypertussive Responses in Rabbits and Guinea Pigs . J Pharmacol Exp Ther. 2016;357: 73–83. doi:10.1124/jpet.115.230227

7. When performing BAL, the volume of HEPES solution repeatedly administered is usually re-calculated to body weight of animals. Why you did administer uniform amount of the solution? Was the bw of rabbit equally distributed among the groups?

Answer:

The body weight of rabbits was effectively equally distributed among groups corresponding to 3.222 ± 0.574 kg in control group and 3.210 ± 0.417 kg. in OVA group.

Moreover, the total cell concentration of BAL was determined in order to standardize the number of cell used to identify cell populations (250,00 cells was spun in a cytospin).

8. Latin words should be written in Italics (e.g. rectus abdominis).

Answer:

The text has been modified as follow

M&M:

 Animal preparation …..The electromyographic activity of the rectus abdominis muscle was measured by insertion of bipolar insulated fine stainless steel wire electrodes…..

Fig 2. Typical cough reflex to mechanical stimulation of trachea. The increased expiratory flow is associated with a burst of rectus abdominis EMG activity and…

Data Analysis: 

Ventilatory responses to mechanical stimulation of trachea

 The cough response was identified from the change in tidal volume (VT), peak expiratory flow (V’Epeak) and rectus abdominis EMG [27]……..

Ventilatory responses to chemical stimulation of trachea 

Cough response to nebulisation of citric acid for 4 minutes was quantified from the bursts of the rectus abdominis EMG activity….

Fig 3. Typical reflex cough response during chemical stimulation of trachea quantified from the bursts of the rectus abdominis EMG activity.

Immune-driven inflammation model

 ….Although, this procedure not entirely reflect the inflammation profile of asthma [31], increased eosinophils in BAL and lung tissue have been attested in different animals models i.e. rabbit..

Modulation of chemically induced cough

However, it has been shown that ASIC transcript levels in sensory neurons were increased in inflammatory conditions in vivo [39] and that proinflammatory mediators such as nitric oxide…

….filaments was measured in vitro using a preparation consisting from isolated tracheal segments, vagus nerves and afferent ganglions (Riccio, Myers et al. 1996).

Limitations of the study

…Therefore, we suppose that in our study, the great majority of expulsive events provoked by inhalation of citric acid and quantified from the bursts of the rectus abdominis EMG activity were cough reflexes….

9. In the discussion should be mentioned, why only sensitised animals where used and were not confirmed with non-sensitized animals (e.g. eosinophils).

Answer:

The discussion was modified as follow and 2 supporting information documents were added and cited in the text

 Immune-driven inflammation model

…………. The study protocol was modified to take into consideration both the 3Rs principle in animal experimentation and the efficiency of inflammation induction. In accordance with the 3Rs principle, it has been chosen to reduce the total number of animals in the study. We have therefore used only sensitized but not exposed rabbits as a control and omitted the use of non-sensitized rabbits as another control. In the study of Zschauer et al. (1999) [31] the same profile of cell was observed between control rabbits (not sensitized and saline challenged) and sensitized rabbits challenged with OVA. Moreover, cough threshold observed in control group of current study (150 msec (75 - 525 msec) was similar to that observed in another study using identical protocol of mechanical cough provocation (300 msec (150 - 600 msec) (Varechova, Demoulin et al. 2015). Responses of sensitized rabbits to dermal tests showed a relatively broad induration size range and as far as possible, rabbits with smallest indurations were intended to control group i.e. not exposed to OVA aerosols and those with largest indurations to OVA group (exposed to OVA aerosols) (S1 Fig and S1 Table). As the use of OVA aerosol leads to the development of an antigen tolerance and the quick resolution of inflammation after discontinuing antigen exposures (Stevenson and Birrell 2011) we realized preliminary experiments to determine the optimal aerosol challenge number that allowed us to reproduce some key features of clinical acute asthma, such as early and late-phase bronchoconstriction in response to allergen challenge and the influx of eosinophils into lower respiratory tract resulting in eosinophil airway inflammation. Realization of two OVA challenges in sensitized rabbits, 48 h and 24 h before experiments, provoked significant increase of eosinophils (S1 Table) that was comparable with those in study of Xue and co-workers that used five allergen challenges….

10. The first sentence in the discussion should be more specific for rabbits, as there are available some studies about ovalbumin sensitisation and challenge on cough reflex in guinea pigs 

Answer:

The originality of this study was to investigate the effect of acute immune-driven inflammation associated with allergic challenge on reflex cough in sensitized rabbits as underlining by the reviewer.

The first sentence in the discussion was consequently modified as follow

This study is the first one to investigate the effect of acute immune-driven inflammation associated with allergic challenge on reflex cough in sensitized animals rabbits in vivo.

---

## [Decision Letter · Decision Letter 1]

27 Nov 2019

Modulation of protective reflex cough by acute immune driven inflammation of lower airways in anesthetized rabbits

PONE-D-19-22841R1

Dear Dr. Demoulin-Alexikova,

We are pleased to inform you that your manuscript has been judged scientifically suitable for publication and will be formally accepted for publication once it complies with all outstanding technical requirements.

With kind regards,

Yu Ru Kou, PhD

Academic Editor

PLOS ONE

Additional Editor Comments (optional):

Reviewers' comments:

Reviewer's Responses to Questions

**Comments to the Author**

1. If the authors have adequately addressed your comments raised in a previous round of review and you feel that this manuscript is now acceptable for publication, you may indicate that here to bypass the “Comments to the Author” section, enter your conflict of interest statement in the “Confidential to Editor” section, and submit your "Accept" recommendation.

Reviewer #1: (No Response)

Reviewer #2: All comments have been addressed

2. Is the manuscript technically sound, and do the data support the conclusions?

Reviewer #1: Yes

Reviewer #2: Yes

3. Has the statistical analysis been performed appropriately and rigorously? 

Reviewer #1: Yes

Reviewer #2: Yes

4. Have the authors made all data underlying the findings in their manuscript fully available?

Reviewer #1: Yes

Reviewer #2: Yes

5. Is the manuscript presented in an intelligible fashion and written in standard English?

Reviewer #1: Yes

Reviewer #2: Yes

6. Review Comments to the Author

Reviewer #1: (No Response)

Reviewer #2: Congratuations to the authors and appreciation for answering all my questions and previous concerns. I recommend the manuscript to be published in this journal.

7. PLOS authors have the option to publish the peer review history of their article (what does this mean?). If published, this will include your full peer review and any attached files.

Reviewer #1: No

Reviewer #2: No

---

## [Editor Report · Acceptance letter]

6 Dec 2019

PONE-D-19-22841R1 

Modulation of protective reflex cough by acute immune driven inflammation of lower airways in anesthetized rabbits 

Dear Dr. Demoulin-Alexikova:

I am pleased to inform you that your manuscript has been deemed suitable for publication in PLOS ONE. Congratulations! Your manuscript is now with our production department. 

With kind regards,

on behalf of

Dr. Yu Ru Kou 

Academic Editor

PLOS ONE